# PFT: Phonon Fine-tuning for
# Machine Learned Interatomic Potentials

**Teddy Koker** [1]   **Abhijeet Gangan** [2 3]   **Mit Kotak** [4]   **Jaime Marian** [3]   **Tess Smidt** [1]

## Abstract

Many materials properties depend on higher-order derivatives of the potential energy surface, yet machine learned interatomic potentials (MLIPs) trained with a standard loss on energy, force, and stress errors can exhibit error in curvature, degrading the prediction of vibrational properties. We introduce phonon fine-tuning (PFT), which directly supervises second-order force constants of materials by matching MLIP energy Hessians to DFT-computed force constants from finite displacement phonon calculations. To scale to large supercells, PFT stochastically samples Hessian columns and computes the loss with a single Hessian-vector product. We also use a simple co-training scheme to incorporate upstream data to mitigate catastrophic forgetting. On the MDR Phonon benchmark, PFT improves Nequix MP by 55% on average across phonon thermodynamic properties and achieves state-of-the-art accuracy among models trained on Materials Project trajectories. PFT also generalizes to improve properties beyond second-derivatives, improving thermal conductivity predictions that rely on third-order derivatives of the potential energy.

## 1. Introduction

Many computational materials workflows use density functional theory (DFT) to compute materials properties from first principles. DFT's computational cost has motivated

[1]Department of Electrical Engineering and Computer Science, Massachusetts Institute of Technology, Cambridge, MA, USA [2]Department of Civil and Environmental Engineering, University of California, Los Angeles, CA, USA [3]Department of Materials Science and Engineering, University of California, Los Angeles, CA, USA [4]Center for Computational Science and Engineering, Massachusetts Institute of Technology, Cambridge, MA, USA. Correspondence to: Teddy Koker <tekoker@mit.edu>.

*Proceedings of the $43^{rd}$ International Conference on Machine Learning*, Seoul, South Korea. PMLR 306, 2026. Copyright 2026 by the author(s).

machine learned interatomic potentials (MLIPs) as fast surrogates that can replace or accelerate DFT in large-scale screening and simulation (Yang et al., 2024; Merchant et al., 2023). Accurate predictions of phonon and vibrational properties require an MLIP to match the *curvature* of the DFT potential energy surface (PES), not just energies, forces, and stress. However, existing "universal" MLIPs are typically trained with supervision over energy, force, and stress; this only indirectly constrains second derivatives, leading to errors in the curvature that degrade phonon dispersions and thus various thermodynamic properties (Deng et al., 2025). In this work, we show that Hessian error strongly correlates with error across multiple phonon thermodynamic metrics.

In materials, training directly on curvature is challenging because phonon force constants in periodic crystals are commonly obtained by finite-displacement calculations in supercells; the supercell must be large enough that displaced atoms do not interact with periodic images of themselves, and to capture interactions that extend beyond the primitive cell. These requirements often necessitate hundreds or thousands of atoms, where the $3N \times 3N$ Hessian, or force constant matrix scales quadratically in system size, making full Hessian training infeasible.

We introduce **phonon fine-tuning (PFT)**, a fine-tuning procedure that directly incorporates second-order force constants by matching energy Hessians from the MLIP to DFT-derived force constants (Fig. 1c(iv)). To scale to large supercells, PFT stochastically samples force constant columns and computes the corresponding loss via a single Hessian-vector product, reducing the training step cost from quadratic to linear with respect to the number of atoms. Because of the relative diversity of existing phonon datasets and lack of non-equilibrium geometries, we introduce a simple co-training strategy to mitigate catastrophic forgetting by interleaving upstream pretraining data during fine-tuning.

We evaluate PFT by fine-tuning the Nequix MP foundation model (Koker et al., 2025) and benchmarking on held out calculations from the PBE MDR Phonon benchmark (Togo, 2023a; Loew et al., 2025). At a fraction of the cost of the initial pretraining, PFT reduces property error by 55% on average across phonon properties: maximum phonon frequency, vibrational entropy, Helmholtz free energy, and heat

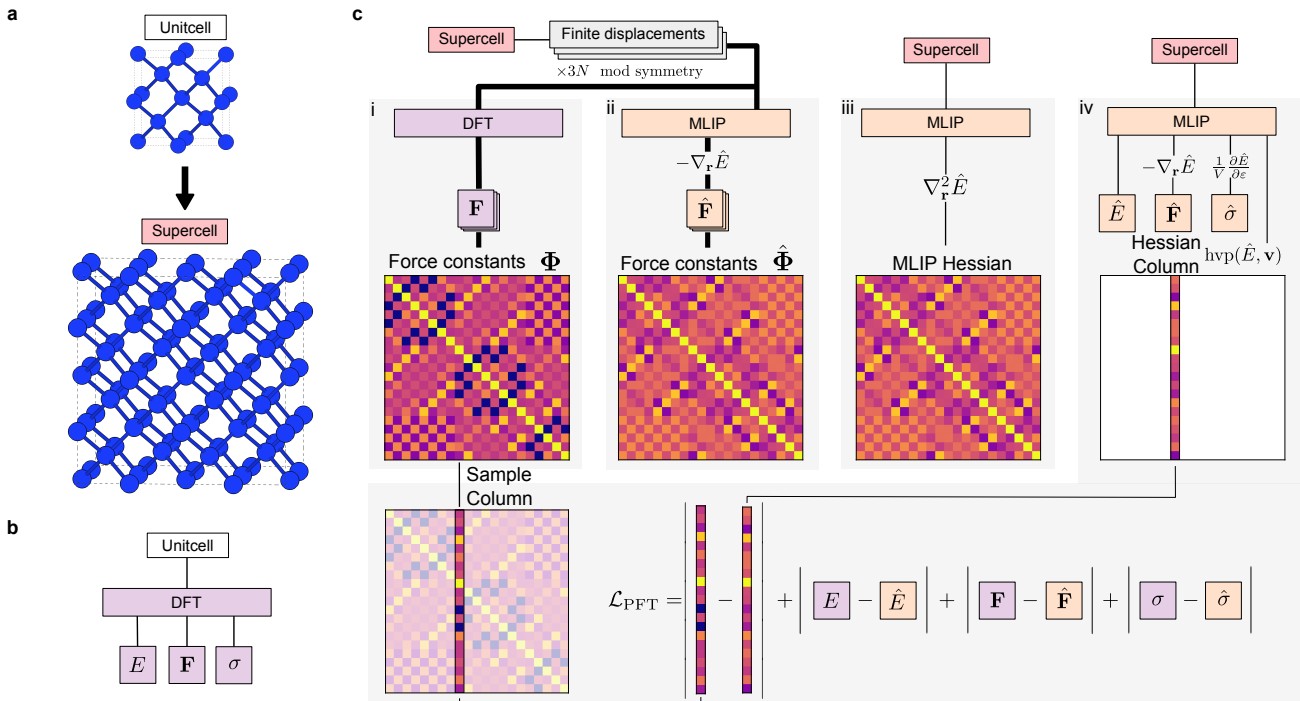

*Figure 1.* **Overview of PFT framework**. **a** Finite-difference calculations rely on the construction of a supercell to obtain force constants from interactions beyond the unitcell. **b** MLIPs are pre-trained on standard unitcell DFT calculations. **c**-i Up to $O(3N)$ atomic displacements are applied to the supercell, with the number reduced by crystal symmetries; forces are computed with DFT, and numerical derivatives yield the force-constant matrix. **c**-ii The same workflow can be used for MLIPs, replacing DFT force calculations with force predictions from the model. **c**-iii Alternatively, the force constants can also be computed as the analytical Hessian of the predicted energy directly. **c**-iv In this work, we use the Hessian-vector product to efficiently compute columns of the Hessian, and align with sampled DFT force constant columns during training. Note the shown force constants are downsampled for clarity.

capacity, achieving state-of-the-art accuracy among models trained on MPtrj (Deng et al., 2023; Jain et al., 2013). Furthermore, we demonstrate that PFT improves generalization to third-order derivatives, improving Matbench Discovery thermal conductivity predictions from 0.446 to 0.307 $\kappa_{\text{SRME}}$, which is also state-of-the-art among MPtrj-trained models. We demonstrate that through co-training, PFT degrades performance by less than 1% on the Matbench Discovery stability classification task, which is otherwise greatly impacted by training on phonon data alone. Lastly, we apply PFT to a stronger base model trained on OMat24 (Barroso-Luque et al., 2024), observing consistent benefits despite substantially more upstream data.

Our contributions are as follows:

- A fine-tuning objective that directly aligns the curvature of the MLIP PES by aligning energy Hessians with DFT-derived force constants.
- A scalable strategy for training on phonons of large supercells by sampling columns of the Hessian, and training with Hessian-vector products.
- A simple co-training recipe that mitigates catastrophic

forgetting by interleaving pre-training data into the fine-tuning procedure.

- Empirical results demonstrating that Hessian error strongly correlates with phonon property error, and that the introduced training objective greatly improves performance on phonon properties while preserving performance on other tasks.

## 2. Background

### 2.1. Machine Learned Interatomic Potentials

Machine learned interatomic potentials (MLIPs) seek to model the Born-Oppenheimer potential energy surface $E(\mathbf{r}, \mathbf{z})$ for atoms with positions $\mathbf{r} \in \mathbb{R}^{3N_a}$, and species $\mathbf{z} \in \mathbb{Z}_+^{N_a}$, where $N_a$ is the number of the atoms in the system. Neural network MLIPs typically build a model for predicted energy $\hat{E}_\theta(\mathbf{r}, \mathbf{z})$ parameterized by weights $\theta$, based on local atomic environments (Unke et al., 2021). These models are trained on quantum chemistry calculations such as density functional theory (DFT) (Hohenberg & Kohn, 1964).

Recent work has demonstrated the success of so-called *universal* MLIPs, which seek to model the PES across a broad range of chemistries and geometries by training on vast DFT databases (Chen & Ong, 2022; Batatia et al., 2025a; Wood et al., 2025). These models offer the promise of acting as a surrogate to otherwise computationally expensive quantum chemistry calculations, enabling higher-throughput computational materials workflows.

In this work we focus on *energy-conserving* MLIPs, where forces and stresses are obtained as derivatives of a scalar energy model, which ensures that the curl of the predicted forces are zero by design.

With atom positions $\mathbf{r}$ in a periodic cell of volume $V$, predicted forces are computed as

$$\hat{\mathbf{F}}_a = -\nabla_{\mathbf{r}_a} \hat{E}(\mathbf{r}) \tag{1}$$

Predicted stress on the lattice is computed as the derivative of energy with respect to symmetric strain tensor $\varepsilon$:

$$\hat{\sigma}_{ij} = \frac{1}{V} \left. \frac{\partial \hat{E}(\mathbf{r})}{\partial \varepsilon_{ij}} \right|_{\varepsilon=0} \tag{2}$$

These quantities are convenient to compute with automatic differentiation (AD) from $\hat{E}$. Note that for the remainder of the paper we have dropped the dependence on atomic species $\mathbf{z}$ for clarity.

MLIPs for periodic systems are typically trained using a loss over energies, forces, and stresses computed from DFT. This is constructed as a weighted sum of errors over each term:

$$\mathcal{L}_{\text{EFS}} = \lambda_E \mathcal{L}_E + \lambda_F \mathcal{L}_F + \lambda_\sigma \mathcal{L}_\sigma \tag{3}$$

with individual loss terms

$$\mathcal{L}_E = \left| \frac{\hat{E}}{N_a} - \frac{E}{N_a} \right| \qquad \mathcal{L}_F = \frac{1}{N_a} \sum_{a=1}^{N_a} \left\| \hat{\mathbf{F}}_a - \mathbf{F}_a \right\|_2 \tag{4}$$

$$\mathcal{L}_\sigma = \frac{1}{9} \sum_{i=1}^{3} \sum_{j=1}^{3} |\hat{\sigma}_{ij} - \sigma_{ij}| \tag{5}$$

where $E$, $\mathbf{F}$, and $\sigma$ denote DFT-computed energy, force, and stress respectively, and coefficients $\lambda$ are tunable hyperparameters to allow weighting of different quantities. Universal potentials are often trained on databases of relaxation trajectories (Deng et al., 2023; Jain et al., 2013; Schmidt et al., 2024) or by perturbing equilibrium structures to sample more of the PES (Barroso-Luque et al., 2024; Kaplan et al., 2025). Prior work has found that due to the bias of data towards equilibrium structures, there often exists a softening in the curvature of the PES (Deng et al., 2025).

## 2.2. Harmonic Phonons and Vibrational Properties

Phonons arise from small lattice vibrations around a local minimum of the PES, and their spectra and scattering control many materials properties of interest, including thermal conductivity, thermal expansion, and heat capacity (Ziman, 2001). They also dictate dynamic stability and, via electron-phonon coupling, can enable superconductivity (Giustino, 2017).

Phonon frequencies are calculated from the eigenvalues of the dynamical matrix, which is constructed by mass-weighting the real-space force constants $\Phi$, and applying a lattice Fourier transform that introduces the wavevector $\mathbf{k}$ dependence. The force constants are defined as the second derivative of the PES with respect to atomic positions:

$$\Phi_{aibj} = \frac{\partial^2 E}{\partial r_{a,i} \partial r_{b,j}} \tag{6}$$

where $i$, $j$ are Cartesian indices, and $a$, $b$ are atom indices.

## 2.3. Phonon Calculations

Phonon calculations are usually conducted with DFT in one of two ways: 1) using density functional perturbation theory (DFPT), or 2) using DFT with finite displacements. The latter is more common due to its generality across commonly used functionals (Miranda).

With finite displacement, second-order force constants are approximated with

$$\Phi_{aibj} \simeq -\frac{F_{b,j}(\Delta r_{a,i}) - F_{b,j}}{\Delta r_{a,i}} \tag{7}$$

$F_{b,j}(\Delta r_{a,i})$ are forces where atom $a$ is displaced in direction $i$, and generally $F_{b,j} = 0$ due to the atom positions being at equilibrium. This approximation works well due to the smoothness of DFT under small displacements, commonly 0.01 Å for these calculations. The number of finite-displacement calculations (3N) can typically be significantly reduced by leveraging the symmetry of the crystal (Togo et al., 2023).

It is important to note that phonon calculations using finite displacement require large supercells for calculations to avoid self-interaction with the displaced atom, and correctly model interatomic force constants that extend past the unit cell (Fig. 1a).

MLIPs can be used for phonon calculations in the same way as DFT, constructing force constants with force calculations under finite displacement (Fig. 1c(i-ii)), which has shown success in predicting second-order (Loew et al., 2025) and third-order (Póta et al., 2024) phonon properties.

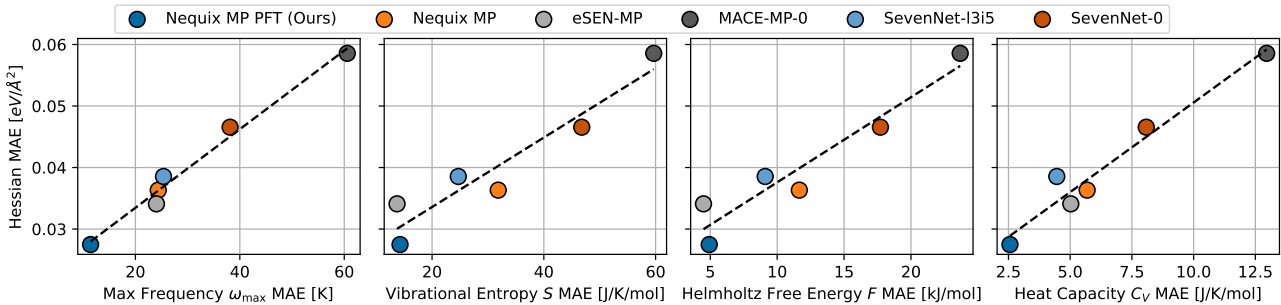

*Figure 2.* **Hessian error vs. phonon properties.** Error in the Hessians on the test subset of the MDR Phonon data are plotted against max phonon frequency, vibrational entropy, Helmholtz free energy, and heat capacity errors for several foundation models trained on MPtrj. Reduced Hessian errors correlate with improved property prediction.

## 3. Methodology

### 3.1. Hessian Error and Vibrational Properties

Because phonon spectra and derived vibrational properties are functions of the second-order force constants, models must match the curvature of the PES to achieve accurate phonon predictions. Using the finite displacement method above, we calculate the error in the second-order force constants, or Hessians of models trained on MPtrj (Deng et al., 2023; Jain et al., 2013), evaluating them on a portion of the force constants and phonon properties from the PBE MDR phonon dataset (Loew et al., 2025; Togo, 2023a). Figure 2 shows that lower Hessian error correlates with lower error on multiple thermodynamic properties. This motivates that training directly on second-order force constants results in more accurate predictions of these properties.

### 3.2. Phonon Fine-tuning (PFT)

We propose phonon fine-tuning (PFT), a method for fine-tuning MLIPs directly on DFT-computed force constants. $\mathcal{L}_{\text{PFT}}$ is a four term loss function, with terms that minimize error in the energy, force, stress, and force constants:

$$\mathcal{L}_{\text{PFT}} = \lambda_E \mathcal{L}_E + \lambda_F \mathcal{L}_F + \lambda_\sigma \mathcal{L}_\sigma + \lambda_\Phi \mathcal{L}_\Phi \quad (8)$$

where we define

$$\mathcal{L}_\Phi = \frac{1}{3N_a} \sum_{a=1}^{N_a} \sum_{i=1}^{3} \mathbb{E}_{\substack{b \sim \mathcal{U}[1,N_a] \\ j \sim \mathcal{U}[1,3]}} \left| \frac{\partial^2 \hat{E}}{\partial r_{a,i} \, \partial r_{b,j}} - \Phi_{aibj} \right| \quad (9)$$

where $N_a$ is the number of atoms in a system. Note that again the batch dimension is omitted for readability. $\hat{E}$ is the predicted potential energy from the neural network; the first three terms follow the standard EFS loss from Eq. (3) with an MAE loss on energy and stress and a $\ell_2$ loss on forces. The second-order force constants are predicted an-

alytically as the Hessian of the energy with respect to two atom positions.

To improve computational efficiency and enable training on large supercells, we uniformly sample one atom and Cartesian direction for each structure in the batch, effectively selecting one column of the Hessian. This requires only a single Hessian-vector product (see next section) to compute the loss across the whole batch, while effectively still training on the full Hessian in expectation. This reduces the computational complexity of a training step from $O(N^2)$ to $O(N)$ for $N$ number of atoms. Furthermore, by using a symmetry-aware E(3)-equivariant architecture, many Hessian elements are redundant due to the high symmetry of the force constants of a crystal structure (Califano et al., 1981). More consideration may be necessary with regards to sampling and data augmentation if non-equivariant architectures are used.

Since the DFT phonon calculations themselves consist of energy, force, and stress calculations under displacements of atoms, it is reasonable to assume that one could simply fine-tune models directly on this displacement data, as it contains sufficient information for constructing the full Hessian. Empirically we find this not to be the case, and observe that directly fine-tuning on the phonon displacements results in a significantly *worse* Hessian error (Fig. 3, Table 1). This suggests that the standard practice of EFS training on DFT calculations of rattled, or perturbed structures (Barroso-Luque et al., 2024; Kaplan et al., 2025) may not be sufficient for correctly modeling the PES curvature.

### 3.3. Efficient Computation via Hessian-Vector Product

PFT requires computing a subset of predicted second derivatives to compare to DFT-computed force constants for large supercells. For a structure with atomic positions $\mathbf{r} = (\mathbf{r}_1, ... \mathbf{r}_{N_a}) \in \mathbb{R}^{3N}$ we form a selection vector $\mathbf{v}$ that is composed of zeros everywhere except for the index cor-

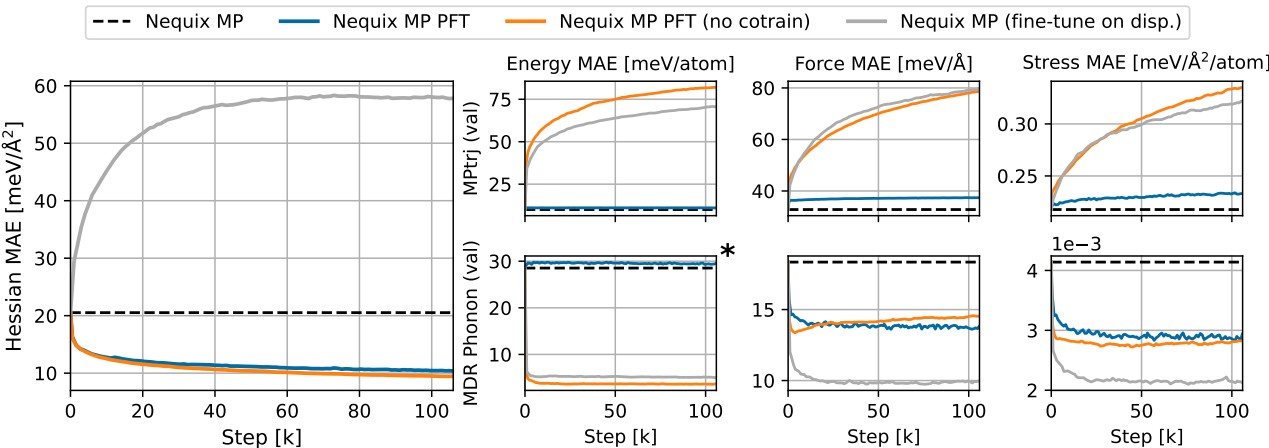

*Figure 3.* **Training ablation**. The left figure shows Hessian error on the MDR Phonon validation set, and the remaining figures show energy, force, and stress errors on the MPtrj validation set (top) and the MDR Phonon validation set (bottom). We compare phonon fine-tuning with and without co-training on MPtrj, as well as directly fine-tuning energy, force, and stress on the phonon displacement calculations. Co-training mostly mitigates degradation of MPtrj performance at only a slight increase in Hessian error. Training directly on displacements significantly worsens Hessian error over the base model. *We note that larger energy errors on MDR Phonon data are likely due to a mismatch in energies between MPtrj and MDR Phonon, see Sec. A.2.

responding to $\mathbf{r}_{b,j}$, the $j$-th Cartesian component of atom $b$, where it is one. The corresponding Hessian column is then

$$\nabla_{\mathbf{r}}^2 \hat{E}(\mathbf{r})\mathbf{v} = \nabla_{\mathbf{r}}\left(\nabla_{\mathbf{r}}\hat{E}(\mathbf{r})^{\top}\mathbf{v}\right) \qquad (10)$$

which can be computed without materializing the full Hessian by using a Hessian-vector product (HVP), using a forward-mode Jacobian-vector product (JVP) through a reverse-mode gradient (Pearlmutter, 1994). In JAX (Bradbury et al., 2018), this is implemented concisely as:

```
def hvp(energy, pos, v):
    return jax.jvp(
        jax.grad(energy),
        (pos,),
        (v,)
    )[1]
```

Here `energy` corresponds to model energy $\hat{E}$, `pos` are the atomic positions $\mathbf{r}$, and `v` is the direction vector $\mathbf{v}$. In practice, Hessian columns across a full batch of structures are computed by forming a single graph of atom positions where each structure is not connected, concatenating the sampled $\mathbf{v}$ for each structure into a single vector, and computing the HVP with respect to the sum of energies across all structures. This facilitates calculation of the loss with a single HVP call, and enables training on GPUs that would otherwise not have enough memory for a single Hessian calculation, especially for the large supercells that are needed for accurate phonon calculations (see Sec. A.1). During training, this results in a "triple-backward" as gradient-based optimization requires a derivative of the HVP with respect to model weights.

### 3.4. Co-training

While fine-tuning models on phonon data will more closely align the curvature of the PES with that of DFT, this procedure may lead to catastrophic forgetting (French, 1999), where the fine-tuning procedure affects the performance of the model on its original upstream training data. This may be an issue as phonon calculations are always done at equilibrium, which can cause forgetting for out-of-distribution non-equilibrium configurations. Furthermore, the relative quantity and diversity of the phonon dataset may be less than the original pretraining or fine-tuning datasets (such as the case in our experimental setting).

We propose a simple solution to this by alternating training steps between PFT steps and a typical energy/force/stress (EFS) loss on the original upstream dataset, as outlined in Algorithm 1. The ratio of EFS on the upstream dataset to PFT steps on the phonon dataset $K$ can be tuned by monitoring validation datasets for both the upstream and phonon datasets during training. As shown in Fig. 3, we demonstrate that without co-training, PFT causes energy, force, and stress errors to deviate quite significantly on the upstream validation dataset. The introduction of co-training, however, mostly eliminates this deviation with only a small reduction in Hessian MAE.

### 3.5. Analytical Property Prediction

As discussed, most existing methods for evaluating MLIPs on higher order derivatives of the PES do so with finite displacement (Loew et al., 2025; Póta et al., 2024). This

---

**Algorithm 1** PFT training with co-training

---

**Require:** Phonon dataset $\mathcal{D}_{\text{phonon}}$, upstream dataset $\mathcal{D}_{\text{up}}$, co-training ratio $K$, model $\hat{E}_\theta$

1: **for** batch in $\mathcal{D}_{\text{phonon}}$ **do**
2:     Sample $(b, j) \sim \mathcal{U}([1, N_a] \times [1, 3])$ per structure
3:     Update $\theta$ using $\mathcal{L}_{\text{PFT}}$ (Eq. 8)
4:     **for** $k = 1$ to $K$ **do**
5:         Sample batch from $\mathcal{D}_{\text{up}}$
6:         Update $\theta$ using $\mathcal{L}_{\text{EFS}}$ (Eq. 3)
7:     **end for**
8: **end for**

---

is typically accurate if the smoothness and continuity of the PES is taken into account when designing the neural network architecture; however, it does introduce an additional hyper-parameter in the form of displacement distance, which can subtly affect results (Fu et al., 2025).

Leveraging the HVP, we can compute the full force constants, see Eq. (6), by iterating over Hessian columns, parallelizing HVP calls until GPU memory is saturated. While in theory this computation may need to be conducted on a graph that exceeds the receptive field of the neural network (Fang et al., 2024), simply using the same supercell as the DFT calculation will ensure we can sufficiently predict the same phonon modes as the ground truth. In practice we find that we achieve near identical results through analytical force constant prediction and finite displacement, which we show in Table 1.

## 4. Experiments

### 4.1. Training

For all experiments, we start with the Nequix MP (Koker et al., 2025), whose JAX (Bradbury et al., 2018) implementation enables convenient auto-differentiation through the Hessian terms of the loss function. Nequix MP was trained on MPtrj (Deng et al., 2023), a dataset of relaxation trajectories from the v2022.10.28 release of Materials Project (Jain et al., 2013) consisting of 145,923 unique materials. MPtrj will serve as the upstream dataset $\mathcal{D}_{\text{up}}$.

For phonon calculation data $\mathcal{D}_{\text{phonon}}$, we use MDR Phonon calculation database (Togo, 2023a), which was recalculated by (Loew et al., 2025) using the PBE exchange correlation functional in order to match the settings of the Materials Project data. These calculations were conducted using finite displacement (see Sec. 2.3), and contain the original energy/force/stress calculations at each displacement. While the phonon data itself does not contain force, or stress labels needed for the PFT loss function (Eq. 8), we assume the force and stress labels are zero due to the strict structural relaxation procedure done prior to the phonon calculation.

The energy label is computed by multiplying the provided unitcell energy by the number of repetitions within the supercell used for calculations. Of the 9,959 materials within the dataset, all but 91 exist in MPtrj based on mp-id. These 91 materials along with randomly selected calculations from the remaining data are used as a test set of 1,000 materials, with the rest of the data split into training/validation subsets with proportion 95/5. This ensures any benefit we see from PFT is due to the additional phonon information, and not novel chemistry or geometries. The training set then contains 8,510 structures derived from 301,414 displacement DFT calculations. We do note there is a discrepancy between the energies of materials in phonon data and MPtrj, which we detail in Sec. A.2.

We perform PFT both with ($K = 4$) and without ($K = 0$) co-training for 200 epochs on the MDR Phonon data. With co-training, this requires about 35 A100 hours, with about 15 A100 hours for PFT without co-training — significantly fewer than the 100 A100 hours needed to train the base Nequix MP model and a small fraction of other competitive MPtrj-trained MLIPs (Koker et al., 2025). To evaluate robustness to base model quality and scale of pre-training data, we pre-train a new base model, Nequix OAM (see Sec. A.3.1), and apply PFT using the same procedure as Nequix MP. For details on other hyper-parameters and their selection see Sec. A.3.

In order to demonstrate that PFT is model-agnostic, we also apply PFT to MACE-MP-0 (Batatia et al., 2025a), a widely used foundation model also trained on MPtrj. We apply PFT without co-training using the same hyperparameters as Nequix MP. The full fine-tuning procedure requires 30 A100 hours, just over 1% of the original pre-training cost.

### 4.2. Phonon Properties

*Table 1.* Evaluation of models on held-out MDR Phonon data. Metrics are MAE of maximum phonon frequency $\omega_{\text{max}}$ (K), vibrational entropy $S$ (J/K/mol), Helmholtz free energy $F$ (kJ/mol) and heat capacity at constant volume $C_V$ (J/K/mol). [†]Additional upstream data (Sec. A.3.1).

| Model | $\omega_{\text{max}}$ | $S$ | $F$ | $C_V$ |
|---|---|---|---|---|
| MACE-MP-0 | 61 | 60 | 24 | 13 |
| SevenNet-0 | 38 | 47 | 18 | 8 |
| Nequix MP | 24 | 32 | 12 | 6 |
| SevenNet-l3i5 | 25 | 25 | 9 | 4 |
| eSEN-MP | 24 | 14 | **4** | 5 |
| MACE-MP-0 PFT (no cotrain) | 19 | 14 | **4** | 3 |
| Nequix MP (fine-tune on disp.) | 182 | 143 | 59 | 30 |
| Nequix MP PFT | 12 | 14 | 5 | 3 |
| Nequix MP PFT (autodiff) | 12 | 14 | 5 | 3 |
| Nequix MP PFT (no cotrain) | **10** | **11** | **4** | **2** |
| Nequix OAM[†] | 17 | 18 | 7 | 4 |
| Nequix OAM[†] PFT | **9** | **9** | **3** | **2** |

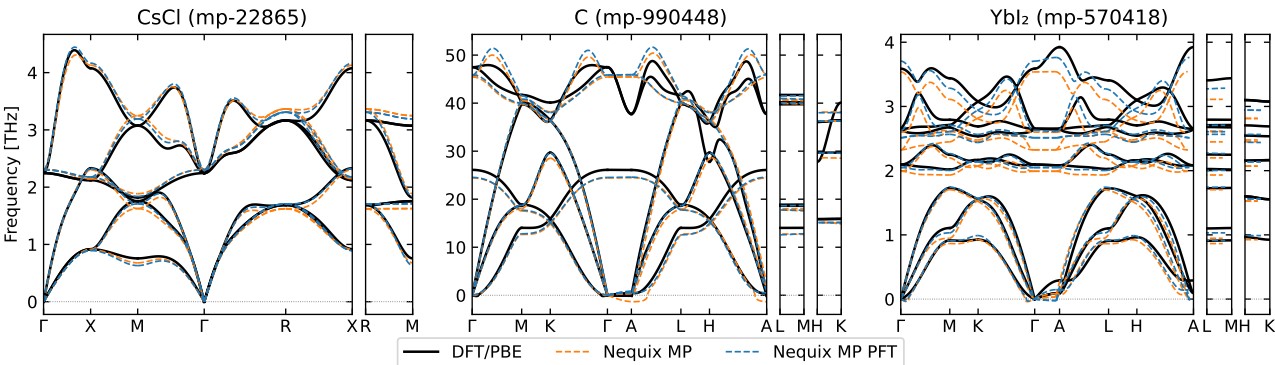

*Figure 4.* **Phonon band structures**. For ease of visualization, we display the phonon band structure for the three materials in the test split of MDR Phonon with the fewest number of atoms. In the case of carbon structure (mp-990448), it results in a dynamically stable structure similar to DFT while the non-PFT model shows an imaginary mode around the $q$-point $A$. Overall, we find that PFT generally produces bands with closer alignment to those of the DFT ground truth. More phonon band structures are provided in Fig. A.5.

Models are evaluated on the 1,000 material test subset of phonon calculations by following the same procedure as Loew et al. (2025), first performing a structural relaxation, and then running phonon calculations with finite displacement using `phonopy` (Togo et al., 2023). Thermodynamic properties are calculated at 300K. We re-run several competitive MPtrj-trained models (Batatia et al., 2025a; Park et al., 2024; Koker et al., 2025; Fu et al., 2025) on the subset. For Nequix MP PFT, we compute force constants with both finite displacement, and analytically via AD.

Table 1 shows the errors of each model on phonon related properties maximum phonon frequency, vibrational entropy, Helmholtz free energy, and heat capacity at constant volume. From the Nequix MP base model, PFT results in an average reduction in MAE of 55%, and achieves state-of-the-art error across all metrics for models trained on MPtrj materials despite being the smallest model. We find no significant differences between properties computed from forces constants obtained via finite displacement or AD, although the PFT model without co-training demonstrates a slight improvement in MAE. Applying PFT to Nequix OAM yields a similar average relative improvement of 50%. Notably, the MP PFT model exceeds the performance of the OAM base model, *despite being trained with approximately two orders of magnitude fewer DFT calculations*.

Lastly, Fig. 4 shows several computed phonon band structures before and after PFT. As is to be expected, we find the PFT leads to band structures that more closely align with the DFT computed band structures.

### 4.3. Third-order Force Constants

While PFT directly supervises second-order force constants, we investigate whether the improved representation of the PES curvature generalizes to third-order force constants, or the third derivative of the energy, which are not explicitly trained on. Just as with second-order force constants, third-order force constants can be calculated with finite displacement, leveraging symmetry to reduce the number of computations (Togo, 2023b). We extracted the displacement data from (Togo, 2025), computed the ground truth DFT third-order force constants, $\Phi^{(3)}$, and then computed the MAE for each model before and after PFT. This is shown in Table 2. We find that with all three base models (Nequix MP, MACE-MP-0, and Nequix OAM), there is a 20-30% reduction in MAE of the third-order force constants after applying PFT.

*Table 2.* MAE of third-order force constants $\Phi^{(3)}$, measured in meV/Å³.

| Model | MAE($\Phi^{(3)}$) |
| --- | --- |
| Nequix MP | 10.52 |
| Nequix MP PFT (no cotrain) | **7.46** |
| Nequix MP PFT | 8.35 |
| MACE-MP-0 | 11.41 |
| MACE-MP-0 PFT (no cotrain) | **7.86** |
| Nequix OAM | 6.46 |
| Nequix OAM PFT | **5.13** |

### 4.4. Thermal Conductivity

To test whether improvements in third-order force constants translate to downstream properties, we evaluate the model on thermal conductivity, a task from the Matbench Discovery (Riebesell et al., 2025) benchmark. Thermal conductivity $\kappa$ is a function of both second and third-order force constants (Póta et al., 2024).

We follow the procedure used by Matbench Discovery

*Table 3.* Matbench Discovery "compliant" leaderboard for thermal conductivity, measured in symmetric relative mean error in predicted phonon mode contributions to thermal conductivity $\kappa_{\text{SRME}}$. [†]Additional upstream data (Sec. A.3.1).

| Model | Params | $\kappa_{\text{SRME}} \downarrow$ |
|---|---|---|
| ORB v2 MPtrj | 25.2M | 1.725 |
| eqV2 S DeNS | 31.2M | 1.676 |
| MatRIS v0.5.0 MPtrj | 5.83M | 0.865 |
| MACE-MP-0 | 4.69M | 0.682 |
| DPA-3.1-MPtrj | 4.81M | 0.650 |
| HIENet | 7.51M | 0.642 |
| SevenNet-l3i5 | 1.17M | 0.550 |
| GRACE-2L-MPtrj | 15.3M | 0.525 |
| Nequip-MP-L | 9.6M | 0.452 |
| Nequix MP | 708K | 0.446 |
| Eqnorm MPtrj | 1.31M | 0.408 |
| eSEN-30M-MP | 30.1M | 0.340 |
| MACE-MP-0 PFT (no cotrain) | 4.69M | 0.397 |
| Nequix MP PFT | 708K | 0.307 |
| Nequix MP PFT (no cotrain) | 708K | **0.281** |
| Nequix OAM[†] | 708K | 0.267 |
| Nequix OAM[†] PFT | 708K | **0.198** |

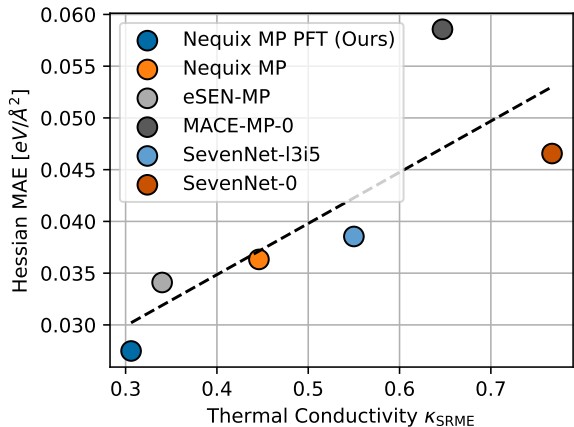

*Figure 5.* **Thermal conductivity vs. Hessian error**. Scatter plot of several models Hessian MAE on the MDR phonon test set vs. symmetric relative mean error in predicted phonon mode contributions to thermal conductivity $\kappa_{\text{SRME}}$. We find there is still a strong trend in between the two despite thermal conductivity using third-order force constants.

(Riebesell et al., 2025; Póta et al., 2024) performing a structural relaxation and then calculation of the third-order force constants with a displacement of 0.03 Å. Results from all of the other MPtrj-trained, or "compliant" models (Neumann et al., 2024; Barroso-Luque et al., 2024; Zhou et al., 2025; Batatia et al., 2025a; Zhang et al., 2025; Yan et al., 2025; Park et al., 2024; Bochkarev et al., 2024; Tan et al., 2025; Chen et al., 2025; Fu et al., 2025) sourced from the Matbench Discovery leaderboard are shown in Table 3.

Similarly to phonon properties we find a reduction in error of 31% from the base Nequix MP model, and state-of-the-art performance for MPtrj-trained models. For the OAM base model, PFT improves error by 26%. In Fig. 6, we show macroscopic conductivity predictions for two zinc blendes, AgI and BeO, showing again that PFT aligns predictions closer to the ground truth DFT result. PFT not only improves second-order vibrational properties, but also generalizes to third-order, or anharmonic, vibrational properties.

### 4.5. Matbench Discovery

The main Matbench Discovery task (Riebesell et al., 2025) consists of a geometry optimization followed by energy prediction to determine material stability. Performance on this task is less influenced by curvature of the PES and largely dictated by energy error, so it highlights any degradation of energy prediction caused by fine-tuning. Table 4 shows the evaluation of the geometry optimization task, measured in root mean squared distance (RMSD) from the DFT-optimized structures as well as the remaining stability

classification metrics from the benchmark.

We find that neither PFT models improve performance over the base model. This may be expected, especially for F1, as the fine-tuning data has no additional energy data from extra chemistries or geometries, so performance of the architecture in this task will be saturated by the original MPtrj training data. However, the co-training is vital for preserving performance on this task; while the non-co-trained PFT model worsens RMSD and F1 by 5% and 60% respectively, co-training reduces this difference to less than a 2% increase in geometry optimization RMSD and less than 1% reduction in any of the other stability classification metrics.

## 5. Related Work

Several works have proposed the use of analytical Hessians with interatomic potentials. Fang et al. (2024) first demonstrated the ability to use AD for phonon prediction through the use of an extended graph, eliminating previously aforementioned issues introduced with self-interaction under finite displacement calculations. They also demonstrate the training of MLIP models on Hessians of small organic molecules. Gangan et al. (2025) demonstrate gradient-based optimization of the classical Stillinger-Weber and EDIP potentials to align with DFT-computed phonon and elastic constant calculations. Amin et al. (2025) introduced a method of model distillation by training smaller MLIP on the analytical Hessians of larger, more accurate models. They also randomly sample columns of the Hessian, but use the Jacobian-vector product of direct-force prediction (i.e. non-energy conserving) models for training.

*Table 4.* Evaluation of models on Matbench Discovery, consisting of geometry optimization, measured in RMSD (Å), coefficient of determination ($R^2$), energy mean absolute error (MAE), discovery acceleration factor (DAF) and stable/unstable material classification F1 and accuracy on unique prototypes. [†]Additional upstream data (Sec. A.3.1).

| Model | RMSD↓ | $R^2$↑ | MAE↓ | Prec↑ | DAF↑ | F1↑ | Acc↑ |
|---|---|---|---|---|---|---|---|
| Nequix MP | **0.085** | 0.782 | **0.044** | 0.681 | 4.455 | **0.751** | **0.914** |
| Nequix MP PFT | 0.087 | **0.784** | **0.044** | 0.685 | 4.479 | 0.748 | **0.914** |
| Nequix MP PFT (no cotrain) | 0.089 | 0.301 | 0.247 | **0.712** | **4.659** | 0.301 | 0.865 |
| Nequix OAM[†] | **0.066** | 0.863 | **0.024** | **0.868** | **5.680** | **0.874** | **0.961** |
| Nequix OAM[†] PFT | 0.067 | **0.865** | 0.025 | 0.863 | 5.648 | 0.864 | 0.958 |

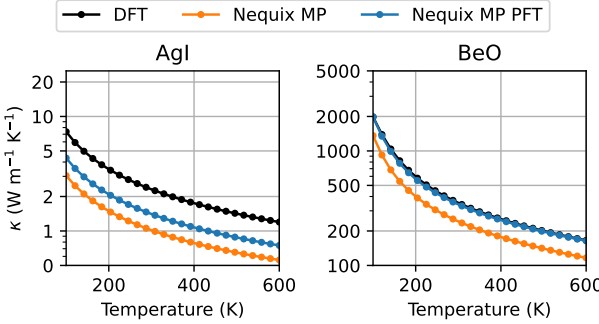

*Figure 6.* **Macroscopic conductivity predictions**. Macroscopic thermal conductivity calculations for zinc blende AgI (left), which exhibits low conductivity, and zinc blende BeO (right), which exhibits high conductivity. DFT calculations are from (Togo, 2025). PFT produces conductivity predictions closer to the ground truth.

Lastly, Burger et al. (2025) proposed the direct prediction of Hessians in small organic molecules, bypassing the need for AD or finite-difference calculations. While this may be beneficial in terms of computational cost, models would not be able to benefit from pre-training on the large quantities of available DFT calculations. Furthermore, tasks such as molecular dynamics, which require lower-order forces, or thermal conductivity which require higher third-order force constants, would be infeasible to conduct.

## 6. Discussion

Phonon fine-tuning (PFT) provides a simple, model agnostic method to improve vibrational and thermal property predictions by directly supervising PES curvature. Across MDR Phonon and Matbench Discovery thermal conductivity, we find that reducing Hessian error aligns phonon-derived observables and can transfer to downstream properties that depend on higher-order energy derivatives. PFT uses force constants already produced by standard DFT phonon workflows, and remains scalable to large supercell via stochastic Hessian column sampling and Hessian-vector products.

We also highlight practical considerations for training on phonon calculations. Training necessitates access to higher-order derivatives through automatic differentiation, e.g., with JAX or PyTorch (see Sec. A.7), which may not be accessible for all universal models. The evaluations in this work consist mainly of materials at or near equilibrium configurations. While we find the PFT models are still able to perform stable MD (Sec. A.6), evaluation of materials MLIP models in off-equilibrium remains an open research direction.

Scaling PFT to larger and more diverse phonon datasets, or with more accurate base models with larger pre-training datasets could improve accuracy and generalization. Beyond force constants, this work could be extended to other DFT-computed higher-order energy derivatives such as elasticity or anharmonic force constants, as well as to properties obtained through experiment.

## Software and Data

We provide the source code used in this work, trained model weights, and preprocessed training data at https://github.com/atomicarchitects/nequix/.

## Acknowledgements

T. K., M. K., and T. S. were supported in part by the National Science Foundation through the AI Research Institutes program Award No. PHY-2019786 (The NSF AI Institute for Artificial Intelligence and Fundamental Interactions, http://iaifi.org/) and Award No. DMR-2433348 (The NSF AI Materials Institute, https://aimi.cornell.edu/), by the Air Force Office of Scientific Research under Award No. FA9550-24-1-0067, as well as by the U.S. Department of Energy, National Nuclear Security Administration under Award No. DE-NA0004266. A. G. and J. M. were supported by DOE Project No. DE-SC0024401. M. K. was also supported by NSF Graduate Research Fellowship program under Grant No. DGE-1745302. This research used resources of the MIT Office of Research Computing and Data, and the National Energy Research Scientific Computing Center (NERSC), a Department of Energy User Facility using NERSC awards ERCAP0033254 and ERCAP0036437.

## Impact Statement

The goal of this work is to advance research in the field of machine learning for computational materials science. There are many potential societal consequences of our work, none of which we feel must be specifically highlighted here.

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

# A. Appendix

## A.1. Full Hessian vs. Stochastic HVP Training

In this section we compare training with stochastic Hessian-vector products (HVP) to training on the full analytical Hessian. Using a very small 90K-parameter model (3 layers, hidden irreps `32x0e + 32x1o + 32x2e`, and a 5 Å radial cutoff), we train from scratch on the force constants of three 64-atom supercells selected to span high, median, and low symmetry (Fig. A.1). Relative to training on the full Hessian, stochastic HVP reduces wall-clock time by approximately $30\times$ and peak GPU memory by $90\times$, while achieving comparable Hessian MAE throughout training. As expected, agreement is strongest for the most symmetric structure, but it remains favorable even in the least symmetric case.

As the time and space complexity of the full Hessian is $O(N^2)$ and HVP is $O(N)$ with respect to number of atoms, this performance gap only grows with more atoms. With larger models and radial cutoffs that are typically used for universal MLIPs and larger supercells needed for accurate phonon calculations, the time and memory usage of training on the full analytical Hessian quickly becomes computationally infeasible.

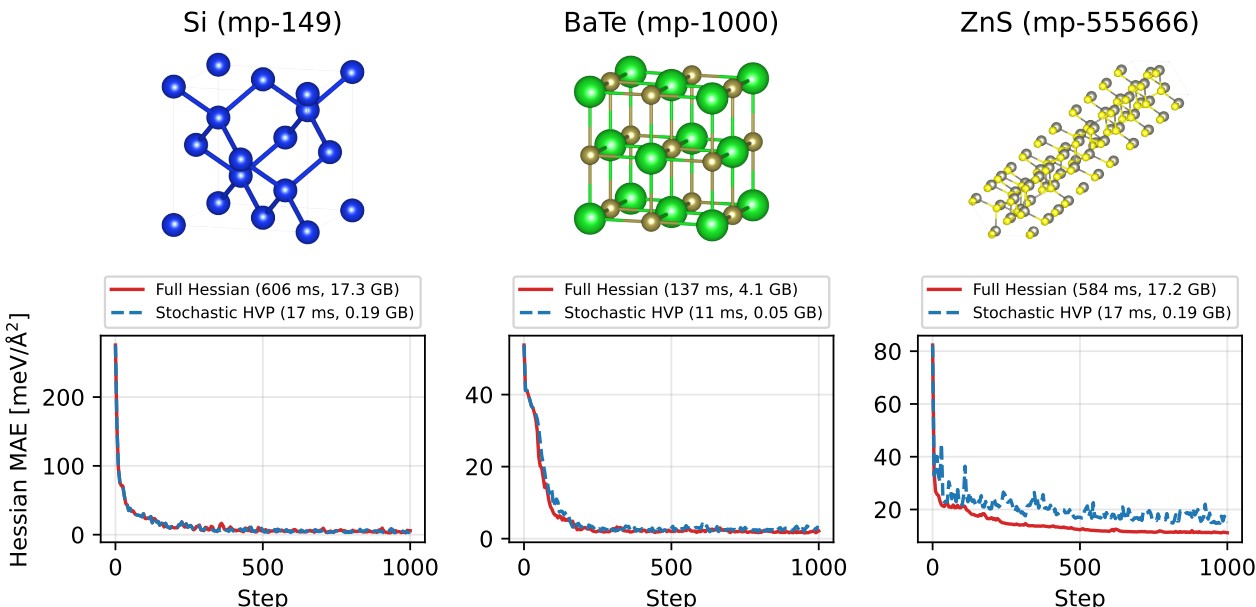

*Figure A.1.* **Full Hessian vs stochastic HVP with Nequix 90K**. We train a small 90K parameter model from scratch on the force constants of individual materials, using both the full Hessian and the stochastic HVP. Each column shows the unit cell, the training time and GPU memory consumption on an NVIDIA RTX A5500, and the full Hessian MAE throughout training.

## A.2. Energy Discrepancy between PBE MDR and MPtrj Datasets

We compare the DFT energies within the PBE recalculation of the MDR Phonon database (Togo, 2023a) by Loew et al. (2025) to those within MPtrj (Deng et al., 2023; Jain et al., 2013). This is performed by selecting the 9868 matching materials across datasets by `mp-id`. For MPtrj we select the final relaxed structure energy, and for both datasets we normalize by number of atoms to offset differences caused by unitcell selection. The energies and the error between them are plotted in Fig. A.2. We find that, while the energies are generally in agreement, there exists a slight shift between the two datasets with a MAE of 31.60 meV/atom. Furthermore there are several outliers with energy differences up to 1.5 eV/atom. We suspect this systemic shift is the cause of the consistent $\approx 30$ meV/atom MAE reported with all interatomic potentials used in Loew et al. (2025), and energy MAE seen in Fig. 3.

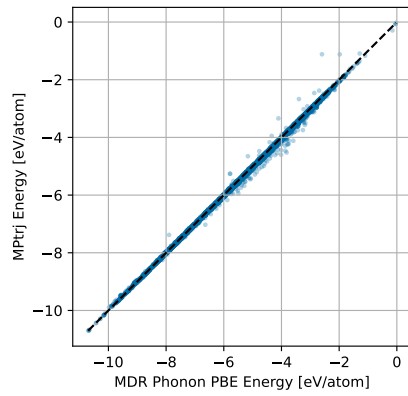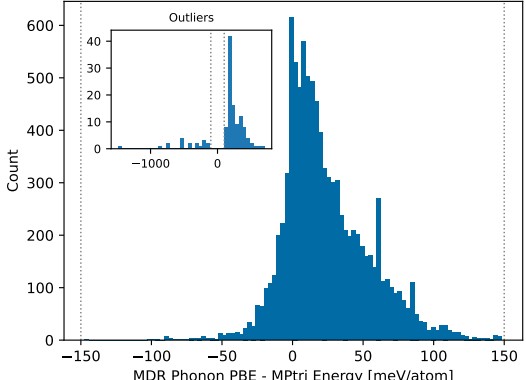

*Figure A.2.* **MDR Phonon and MPtrj energy comparison**. Left: Parity plot of energies from MPtrj (Deng et al., 2023; Jain et al., 2013) vs. PBE MDR Phonon (Togo, 2023a; Loew et al., 2025). Right: Histogram of differences in energy between materials in two datasets. Absolute differences less than 150 meV/atom are shown in the main plot with the remaining outliers in the inset.

## A.3. Hyper-parameters

### A.3.1. TRAINING THE NEQUIX OAM BASE MODEL

Following Fu et al. (2025), the Nequix OAM model is trained by first pretraining on OMat24 (Barroso-Luque et al., 2024), and then fine-tuning on a combination of subsampled Alexandria (sAlex) (Schmidt et al., 2024) and eight copies of MPtrj (Deng et al., 2023; Jain et al., 2013). OMat24 and sAlex contain over 100M and 10M DFT calculations respectively. We use the same training setup as the original Nequix MP model (Koker et al., 2025), using the `torch` backend with OpenEquivariance kernels (Bharadwaj et al., 2025). Changes to hyper-parameters outlined in Table A.1. The OMat pretraining was about 250 A100 hours, and the OAM fine-tuning was an additional 40 A100 hours.

*Table A.1.* Hyper-parameters used for training OMat24 and OAM base models and rationale behind selection.

| Hyper-parameter | Nequix OMat | Nequix OAM | Notes/Rationale |
|---|---|---|---|
| Train data | OMat24 | MPtrj + sAlex | |
| Batch size | 512 | 512 | Maximum for GPU memory ($4\times$A100 80GB). |
| Warmup epochs | 0.1 | 0 | From Nequix MP; no warmup for fine-tuning. |
| # of epochs | 6 | 3 | Models seem to mostly converge at this duration. |

### A.3.2. TRAINING PFT MODELS

Table A.2 shows the hyper-parameters and method for selection for the Nequix PFT models, which are trained using the JAX backend on a single A100 GPU. The MACE-MP-0 PFT (no cotrain) model uses the same setting as Nequix MP PFT (no cotrain), but using the MACE-MP-0 models (Batatia et al., 2025a) with the JAX implementation from FeNNol (Plé et al., 2024).

## A.4. $\lambda_\Phi$ sensitivity

The gain in phonon accuracy is due entirely to the $\mathcal{L}_\Phi$ term of the loss function (Eq. 9), which is weighted in the loss by the coefficient $\lambda_\Phi$. To determine the sensitivity of the PFT procedure to this hyperparameter, we sweep values of 10, 100, and 1000 while keeping the remainder of the loss coefficients and hyperparameters the same. Figure A.3 shows the validation errors of these experiments throughout training. This demonstrates the expected tradeoff: increasing $\lambda_\Phi$ improves Hessian error at the expense of higher errors for the other properties; the inverse occurs for the lower $\lambda_\Phi$. The $\lambda_\Phi = 10$ and $\lambda_\Phi = 1000$ models achieve phonon metrics ($\omega_{\max}/S/F/C_V$) of (17/32/11/4) and (10/9/3/2) respectively, both better than the

*Table A.2.* Hyper-parameters used for PFT models and rationale behind selection.

| Hyper-parameter | Nequix MP PFT | Nequix MP PFT (no co-train) | Nequix OAM PFT | Notes/Rationale |
|---|---|---|---|---|
| Base model | Nequix MP | Nequix MP | Nequix OAM | See Koker et al. (2025) for architecture details. |
| Learning rate | 0.0001 | 0.0001 | 0.0001 | Selected from {0.003, 0.001, 0.0003, 0.0001} based on validation performance early in training. |
| Optimizer | AdamW | AdamW | AdamW | Standard optimizer. |
| Weight decay | 0.001 | 0.001 | 0.001 | From Nequix MP. |
| PFT $\lambda_E$ | 0 | 20 | 0 | From Nequix MP. |
| PFT $\lambda_F$ | 20 | 20 | 20 | From Nequix MP. |
| PFT $\lambda_\sigma$ | 5 | 5 | 5 | From Nequix MP. |
| PFT $\lambda_\Phi$ | 100 | 100 | 100 | See Sec. A.4 |
| Co-train train | MPtrj | n/a | MPtrj + sAlex | Same as training data for base model. |
| Co-train val. | MPtrj | n/a | sAlex | Same as validation data for base model. |
| Co-train $\lambda_E$ | 500 | n/a | 750 | Started with Nequix MP value × 10, increased until co-train validation energy does not diverge over time (see Fig. 3). |
| Co-train $\lambda_F$ | 200 | n/a | 200 | Nequix MP value × 10. |
| Co-train $\lambda_\sigma$ | 50 | n/a | 50 | Nequix MP value × 10. |
| Co-train ratio $K$ | 4 | n/a | 4 | Trade off between training time and overfitting to phonon data; 4 cotraining steps for every phonon step is reasonable. |
| Batch size | 16 | 16 | 16 | Maximum for GPU memory (1×A100 80GB). |
| # of epochs | 200 | 200 | 200 | Not tuned, based on GPU budget. May benefit from longer training, but validation metrics are close to converged at this duration. |

base model, with the latter outperforming the original $\lambda_\Phi = 100$ PFT model (12/14/5/3). This demonstrates that our method is robust to choice in hyperparameter and corroborates our other findings of Hessian error correlating with phonon metrics.

In order to maintain phonon performance without degradation of energy, force, and stress errors, we choose to keep $\lambda_\Phi = 100$ for all other results in this work.

## A.5. Elastic properties

We also benchmark the performance effects of PFT on materials elastic properties, shown in Table A.3. For the benchmark we use the MatCalc Elasticity Calc (Liu et al., 2024) based on the settings as described in the ml-peg package (Kasoar et al., 2025; Batatia et al., 2025b) whereby the elastic tensor and derived properties like the bulk $K$ and shear $G$ modulus are obtained. Voigt-Reuss-Hill (VRH) average is used to report the final values for which we compute the metrics. We observe that the error on shear modulus is lowered for the PFT model while bulk modulus is worsened. Overall, average performance on both moduli is improved for the PFT model. We also report the failure rate for the models for which MatCalc calculations fail to converge.

*Table A.3.* Evaluation of models on Materials Project elastic properties containing 12,122 different materials. Metrics are mean absolute error of bulk modulus $K$ and shear modulus $G$ in units of GPa, correlation score, as well as success rate of computations. To compare the models in a fair way we only consider the materials for which both models return a valid output.

| Model | $K$ MAE | $G$ MAE | $R_K^2$ | $R_G^2$ | Failure |
|---|---|---|---|---|---|
| Nequix MP | **15.03** | 18.16 | **0.89** | 0.44 | 18.37% |
| Nequix MP PFT | 16.49 | **16.41** | 0.87 | **0.54** | 18.21% |

## A.6. Molecular dynamics stability

To verify stability of molecular dynamics (MD) with the introduced models, we follow the procedure from (Fu et al., 2025) and perform 100 picosecond NVE MD simulations at high temperature on five single-vacancy defect transition metal systems selected from TM23 (Owen et al., 2024), measuring the drift in energy. Neither the base, PFT, nor PFT (no cotrain) models showed a measurable energy drift or instability.

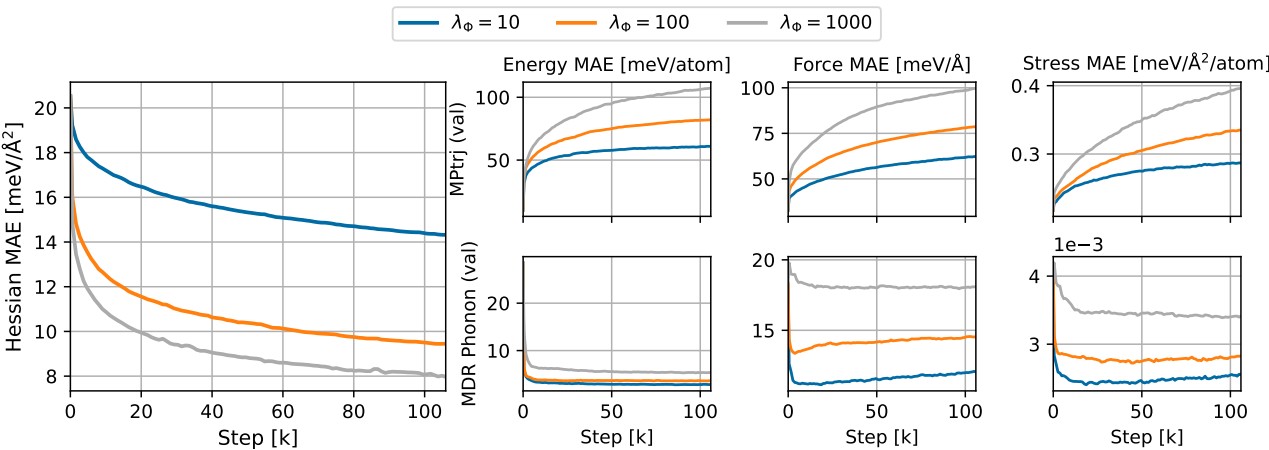

*Figure A.3.* $\lambda_\Phi$ **sensitivity**. Sweep of hyper-parameter $\lambda_\Phi$, the weight of the HVP term in the loss of values (10, 100, 1000) for PFT with the Nequix MP model, with no co-training.

### A.7. PyTorch implementation

PFT can be implemented in PyTorch ([Paszke et al., 2019](#)) by leveraging the functional transforms in `torch.func` and using the same forward-mode JVP through a reverse mode gradient:

```
def hvp(energy, pos, v):
    return torch.func.jvp(torch.func.grad(energy), (pos,), (v,))[1]
```

Again, `energy` corresponds to model energy $\hat{E}$, `pos` are the atomic positions $\mathbf{r}$, and `v` is the direction vector $\mathbf{v}$.

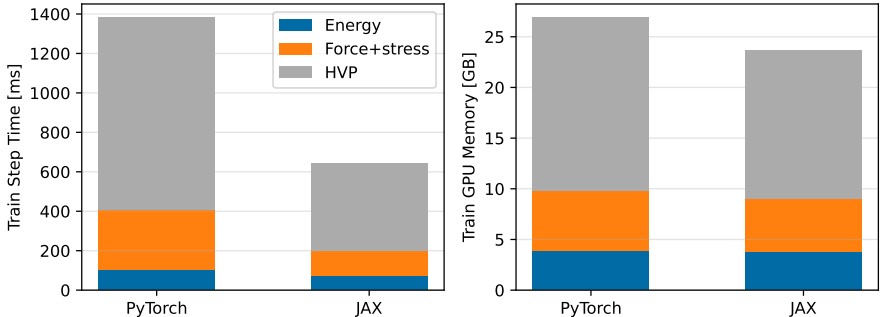

*Figure A.4.* **PyTorch vs JAX training time per step and memory usage**. 64-atom Si supercell with batch size 16 on a A100 GPU. With JAX, the train step with HVP is 3.2× the cost of standard energy/forces/stress step, with 2.6× the memory usage. With PyTorch, the train step with HVP is 3.4× the cost of the standard energy/forces/stress step, with 2.7× the memory usage.

Figure A.4 shows the training speed and memory usage of Nequix in PyTorch and JAX, broken down by energy, force/stress, and HVP loss terms. For sake of comparison, no equivariance kernels such as OpenEquivariance ([Bharadwaj et al., 2025](#)) are used. We find that for both PyTorch and JAX, adding the HVP loss term incurs a roughly 3× time and memory cost, with JAX being significantly faster overall.

## A.8. Phonon band structures

Figure A.5 shows additional examples of band structures calculated with Nequix MP PFT.

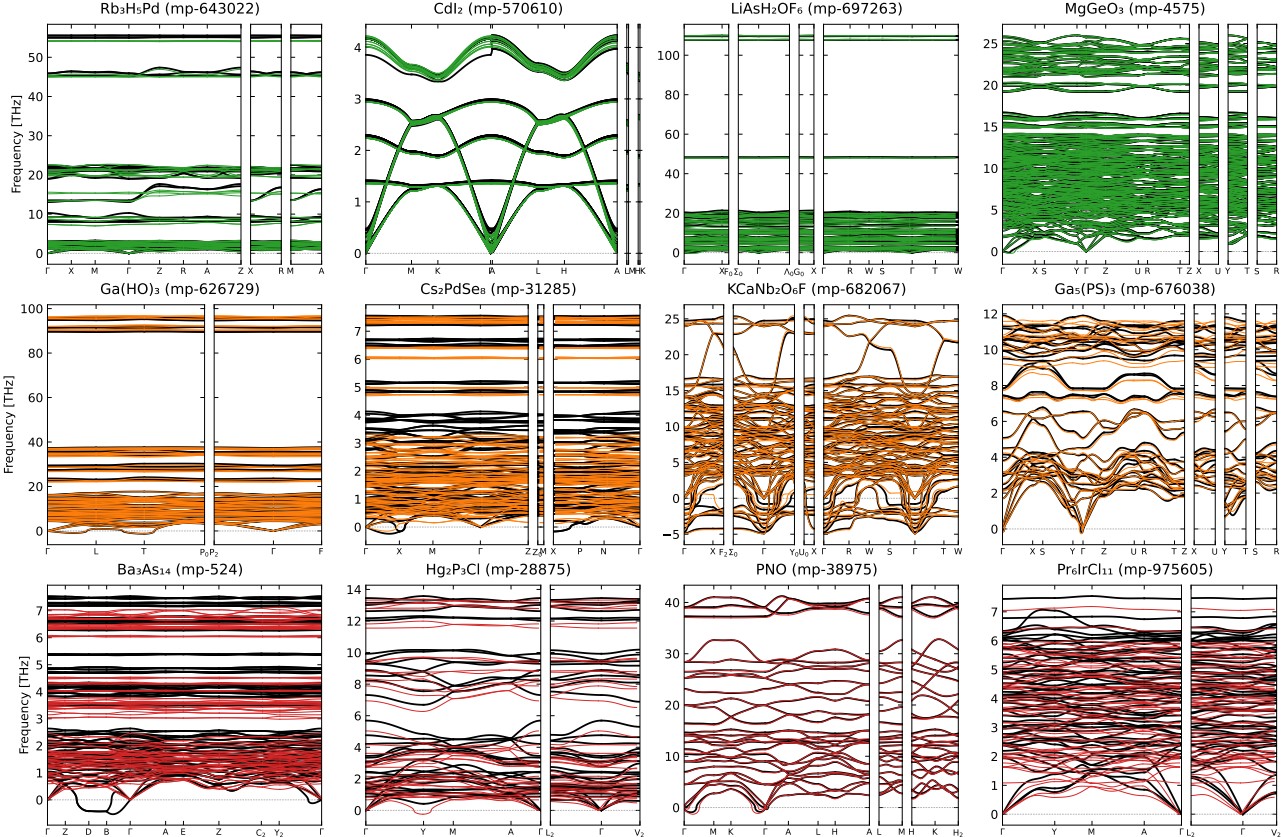

*Figure A.5.* **Phonon band structures with Nequix MP PFT**. Following (Fang et al., 2024), we display predicted band structures (with DFT in black) of four randomly selected materials from each tercile of Hessian error in the PBE MDR phonon test set. The top row shows lowest error, and the bottom row shows highest error materials. Band structures are calculated with `phonopy` (Togo et al., 2023). Note that the number of bands is $3\times$ the number of atoms in the unit cell.

