# OpenReview forum: "PFT: Phonon Fine-tuning for Machine Learned Interatomic Potentials"
_ICML.cc/2026/Conference — ICML 2026 regular_

### Official Review · Reviewer_yQZH · 2026-02-27

**Soundness:** 3
**Presentation:** 4
**Significance:** 3
**Originality:** 3
**Overall Recommendation:** 4
**Confidence:** 5

**Summary:**

This paper proposes phonon fine-tuning to enhance the accuracy of Machine Learned Interatomic Potentials (MLIPs) for properties dependent on higher-order derivatives, such as phonon. The authors note that fine-tuning directly on displacement data (which contains sufficient information for phonon calculation) leads to degraded performance, and find a strong correlation between Hessian (force constant) error and errors across multiple phonon thermodynamic properties. Consequently, they incorporate force constant supervision into the loss function for fine-tuning. As direct Hessian computation is prohibitively expensive, the authors employ stochastic Hessian column sampling and Hessian-vector product (HVP) to reduce computational cost from O(N²) to O(N). A simple co-training strategy is proposed to mitigate catastrophic forgetting. The authors conducted experiments using the JAX-based Nequix MP model with the MPtrj, OAM, and MDR Phonon datasets. Results demonstrate significant improvements on phonon-related tasks including third-order derivative property thermal conductivity, and minimal performance degradation on other tasks (geometry optimization and stable/unstable material classification) when co-training is applied.

**Compliance With Llm Reviewing Policy:**

Affirmed.

**Final Justification:**

Overall, this is a good paper that enables more accurate calculation of phonon spectra. The authors have addressed several concerns through additional experiments. However, the method still requires a significant amount of additional computation and lacks sufficient exploration of non-equilibrium regimes. Therefore, I maintain my original score of weak accept.

**Key Questions For Authors:**

After phonon fine-tuning, how do the errors in energy and forces change? In particular, the properties demonstrated in the paper are strongly related to equilibrium structures (i.e., structural optimization is required before calculation). Can the predictive performance for non-equilibrium structures still be maintained?

**Limitations:**

yes

**Strengths And Weaknesses:**

Strengths:
- The proposed method is intuitive and easy to implement.
- Solutions have been proposed to address the shortcomings of the new method, such as HVP and co-training.
- Experimental validation is comprehensive.

Weaknesses:
- Despite some attempts to reduce computational costs, the computational overhead of the proposed method remains substantial. For instance, the MDR Phonon dataset used in this paper comprises 8,510 structures, but obtaining force-constant information requires 301,414 displacement DFT calculations. Although the fine-tuning time is less than pretraining, it is still considerable. Moreover, the method introduces extra hyperparameters that require optimizing, which may further increase the difficulty of training.

---

> ### Author Rebuttal · Authors · 2026-03-27
>
> Thank you for taking the time to review our work. We have added context and additional experiments which we hope address the weaknesses (W) and questions (Q). If you feel our response addresses your concerns, we kindly ask that you consider whether it merits an increase in score.
>
> **W1: Computational cost of DFT**. While 300 thousand DFT calculations are significant, we would like to highlight that the Nequix MP PFT model, trained on 1.9 million DFT calculations total (1.6 million MPtrj pretraining + 300 thousand phonon), significantly outperforms the Nequix OAM model, trained on 120 million DFT calculations (110 million OMat + 10 million sAlex), in terms of phonon metrics. So while the absolute cost of data generation in this setting is still high, there is a significant benefit in running and training on phonon calculations over conventional methods, which we hope will inform future DFT dataset generation campaigns.
>
> **W2: Computational cost of model training/hyperparameters**. We have added support for OpenEquivariance kernels [1], which brings down the cost of PFT for Nequix from 60 to 15 A100-hours (and from 140 to 35 with co-training). We have also added a new result applying PFT to MACE-MP-0, which took 30 A100-hours to fine-tune, *1% of the original pre-training cost [2].* We achieve comparable gains to the Nequix PFT model, with a 77% average improvement in phonon metrics MAE, and a 38% improvement in $\kappa_\text{SRME}$. These results can be found in response to reviewer mGf7, Revised Table 1 and Revised Table 2. We would also like to highlight that the total aggregate pre-training and fine-tuning cost of Nequix MP PFT is 115 A100-hours, yet achieves greater phonon and thermal conductivity accuracy than all other models trained on MPtrj, including Eqnorm, MACE, and HIENet, all with over 2000 A100-hours [3], and eSEN with over 8000 A100-hours [4]. Given the relative and absolute cost of our method, we do not believe this is a substantial weakness. Lastly, we demonstrate the robustness of hyperparameters with a new study of the effect of $\lambda_\Phi$ (see response to reviewer mGf7 Q1), and with MACE-MP-0 PFT, which achieved similar improvements to the Nequix MP PFT model with identical PFT hyperparameters.
>
> **Q1: Non-equilibrium performance**. We note that "non-equilibrium" can refer to a range of regimes from small displacements around equilibrium (such as phonon calculations or pre-relaxed structures) to highly off-equilibrium configurations, e.g. large distortions. As the upstream MPtrj dataset consists of only near-equilibrium relaxation trajectories, neither the base models nor the PFT model is explicitly trained on highly off-equilibrium geometries.
>
> To assess how well predictive performance is maintained in the near-equilibrium regime, we report energy, force, and stress errors on the MPtrj validation set during fine-tuning (Fig. 3). With co-training, we find these errors remain relatively consistent with the base model. In addition, we evaluate structural relaxation performance (Table 3), and find minimal degradation with respect to the base model.
>
> We agree that evaluating generalization to more strongly non-equilibrium configurations is an important open question that is not well covered by existing benchmarks or training methods. While datasets such as OMat24 have introduced ab initio molecular dynamics trajectories and "rattled" structures for this purpose, the impact of training on this data is an active area of research [5]. We thus view the systematic evaluation of MLIPs in highly off-equilibrium regimes as an important direction for future benchmarking, and will discuss this limitation in the paper.
>
>
> [1] Bharadwaj et al. An Efficient Sparse Kernel Generator for O(3)-Equivariant Deep Networks. Society for Industrial and Applied Mathematics, 2025.
>
> [2] Batatia et al. A foundation model for atomistic materials chemistry. arXiv, 2024.
>
> [3] Koker et al. Training a foundation model for materials on a budget. arXiv, 2025.
>
> [4] Zhou et al. MatRIS: Toward Reliable and Efficient Pretrained Machine Learning Interaction Potentials. arXiv, 2026.
>
> [5] Rhodes et al. Orb-v3: atomistic simulation at scale. arXiv, 2025.

---

> > ### Author Rebuttal · Reviewer_yQZH · 2026-04-02
> >
> > Thank you for your response. The MLIP's predictions for configurations far from equilibrium are crucial for maintaining simulation stability, and I believe this is an issue worth investigating.

---

> > > ### Author Response · Authors · 2026-04-02
> > >
> > > Thank you for the followup. We are glad we have addressed your remaining concerns. We agree that prediction quality far from equilibrium is important for molecular dyanamics (MD) simulation stability.
> > >
> > > To verify stability of MD with the introduced models, we follow the procedure from [1] and perform 100 picosecond NVE MD simulations at high temperature on five single-vacancy defect transition metal systems selected from TM23 [2], measuring the drift in energy. Neither the base, PFT, nor PFT (no cotrain) models showed a measurable energy drift or instability. We will include these results in the paper.
> > >
> > > We agree that a broader study of strongly off-equilibrium configurations and MD stability would be a valuable direction for future work.
> > >
> > > [1] Fu et al. Learning smooth and expressive interatomic potentials for physical property prediction. arXiv, 2025.
> > >
> > > [2] Owen et al. Complexity of many-body interactions in transition metals via machine-learned force fields from the TM23 data set, npj Computational Materials, 2024.

---

### Official Review · Reviewer_w5RP · 2026-03-09

**Soundness:** 3
**Presentation:** 3
**Significance:** 2
**Originality:** 2
**Overall Recommendation:** 3
**Confidence:** 2

**Summary:**

This paper proposes Phonon Fine Tuning (PFT) for pretrained machine-learned interatomic potentials. The main idea is to finetune an energy-conserving MLIP by aligning its energy Hessian with DFT-derived force constants. To reduce the cost on large phonon supercells, the method samples Hessian columns and uses Hessian-vector products for training. The paper also introduces co-training, sampling on pre-training data during fine-tuning to reduce catastrophic forgetting. Experiments on MDR Phonon show improvements on phonon-related properties, and the method also improves thermal conductivity prediction on Matbench Discovery.

**Compliance With Llm Reviewing Policy:**

Affirmed.

**Final Justification:**

The rebuttal clarified the intended scope of the paper and addressed part of my concerns, particularly regarding practicality and the role of co-training.The paper is clearly written, and addresses an important problem related to improving phonon and thermal property prediction for MLIPs. However, my original concerns about the limited general applicability, the supporting rather than central role of co-training, and the limited evidence beyond the target phonon-related metrics are only partially resolved. Overall, I maintain my original assessment and keep my score unchanged.

**Key Questions For Authors:**

1) On co-training. Please discuss the trade-off introduced by co-training and explain more clearly when co-training is necessary and when it is not.
2) On comprehensive evaluation metrics. Please report more complete Matbench Discovery compliant results for Nequix MP PFT, not only κSRME. This would help readers understand whether the gain on thermal conductivity comes with trade-offs on other metrics.
3) On other pretrained MLIPs. Please validate PFT on more pretrained MLIPs, if possible. What Nequix OAM exactly refers to in Table 4. Is this a public model variant, such as L or XL, or a custom internal checkpoint? Ddiscuss the feasibility of larger pretrained models. Although sampled-Hessian design reduces the finetuning cost, the higher-order derivatives may still be expensive as the parameter of pretrained MLIPs grows. The finetuning time and GPU memory usage should be reported.

**Limitations:**

The method is simple, well motivated, and effective on phonon-related tasks. However, the practical scope of the method remains unclear. In addition, the paper’s argument for co-training is not very convincing. The experimental validation is also somewhat narrow.

**Strengths And Weaknesses:**

Strengths
This paper studies an important problem. Many pretrained MLIPs are trained with energy, force, and stress labels only. Such supervision may still be insufficient for learning accurate PES curvature, which is critical for phonons and related properties.

 Weaknesses
1) The applicability of this PFT method. PFT requires DFT force constant matrices. This is a much stronger requirement than standard finetuning. So how broadly PFT can be applied in practice.
2) The role of co-training. The paper presents co-training as an important component because it preserves upstream performance. However, whether preserving performance on the original MPtrj-style distribution is always necessary. If the main goal is downstream specialization, then the foundation model can be viewed as a strong initialization. The results on Table 2 suggest that the no-cotrain variant can perform better on the target downstream task.
3) The model coverage. The experiments are performed on Nequix MP, with additional results on Nequix OAM. This is helpful, but still limited. It remains unclear whether PFT is also effective for other pretrained MLIPs, especially models with large number of parameters.

---

> ### Author Rebuttal · Authors · 2026-03-27
>
> Thank you for the constructive feedback. We have conducted additional experiments to address each weakness (W) and question (Q) below. If you feel our response addresses your concerns, we kindly ask that you consider whether it merits an increase in score.
>
> **W1: Applicability**. While we acknowledge the method requires DFT force constants, there is also high value in achieving accurate phonons, which are necessary for accurate prediction of thermodynamic properties. This is also evident from the prevalence of phonon properties in the two main materials MLIP benchmarks, Matbench Discovery [1] and MDR Phonon [2]. In parallel, there has been significant effort and computational cost in developing new datasets to train MLIPs, including OMat24 [3] which consists of over 110 million DFT calculations. By demonstrating superior phonon performance by fine-tuning on a small fraction of the DFT calculations (300 thousand), our findings will help inform future DFT dataset generation campaigns where training models with accurate phonon properties are desired.
>
> **W2/Q1: Role of co-training**. Co-training preserves performance in settings where non-zero forces exist, or where accurate energy predictions are necessary such as the structural relaxation and stability tasks in Matbench Discovery. There is then a small tradeoff in the reduction of gains due to phonon fine-tuning, however PFT models with co-training still see a significant improvement (55% on average) in phonon metrics with limited impact to the other benchmark metrics (see response to Q2). As structural relaxations are typically done prior to phonon calculations, we believe the co-trained models are likely better suited for production workflows. We will add this commentary to the text.
>
> **Q2: More evaluations**. We have expanded Revised Table 3 below to include the remaining metrics from Matbench Discovery. Note that DAF is simply a scaled Precision, and while it is slightly higher for the non-co-trained model, the F1 is significantly worse. With co-training, all metrics are within 1% of each other. This motivates co-training, as we are able to achieve a significant improvement in phonon and thermal conductivity metrics at a negligible loss in all other metrics. We have also added a new evaluation for third order force constants (see response to reviewer mGf7 Q2).
>
> **W3/Q3: Model variants and cost**.
>    * We have applied PFT to a larger pre-trained model with a different architecture, MACE-MP-0 (4.7M parameters). Using identical PFT hyperparameters to the Nequix MP PFT model, we find that we achieve analogous gains to what was seen with the Nequix model, with a 77% average improvement in phonon metrics MAE, and a 38% improvement in $\kappa_\text{SRME}$ (see response to reviewer mGf7 Table 1 and Table 2). The model took 30 A100 hours to apply PFT, *1% of the original pre-training cost*.
>    * The Nequix OAM base model is discussed in Appendix Section A.3.1. We train a model with the same configuration as Nequix MP from scratch on **O**Mat24 [3], and finetuning on the **A**lexandria and **M**P datasets before applying PFT; the only difference is the upstream data. We will release the model weights with the codebase.
>    * We have also added New Figure A.5 below, which shows the relative speed and memory usage of incorporating the HVP into training. We don't believe the ~3x speed and memory cost prohibits using PFT with most models, and gradient checkpointing can be used to enable much larger models. We will be sure to include these points in the text.
>
> **Revised Table 3**: Evaluation of models on Matbench Discovery, consisting of geometry optimization in RMSD (Å), coefficient of determination ($R^2$), energy mean absolute error (MAE), discovery acceleration factor (DAF) and stable/unstable material classification F1 and accuracy on unique prototypes.
>
> | Model | RMSD | $R^2$| MAE | Prec | DAF | F1 | Acc |
> |---|---|---|---|---|---|---|---|
> | Nequix MP | **0.085** | 0.782 | **0.044** | 0.681 | 4.455 | **0.751** | **0.914** |
> | Nequix MP PFT | 0.087 | **0.784** | **0.044** | 0.685 | 4.479 | 0.748 | **0.914** |
> | Nequix MP PFT (no cotrain) | 0.089 | 0.301 | 0.247 | **0.712** | **4.659** | 0.301 | 0.865 |
> | ----
> | Nequix OAM | **0.066** | 0.863 | **0.024** | **0.868** | **5.680** | **0.874** | **0.961** |
> | Nequix OAM PFT | 0.067 | **0.865** | 0.025 | 0.863 | 5.648 | 0.864 | 0.958 |
>
> **New Figure A.5**: (https://anonymous.4open.science/r/pft-icml/figures/timing.png) Nequix training time per step and memory usage, 64-atom Si supercell with batch size 16 on an A100. The train step with HVP is 3.2x the cost of standard energy/forces/stress step, with 2.6x the memory usage.
>
> [1] Riebesell et al. A framework to evaluate machine learning crystal
> stability predictions. Nature Machine Intelligence, 2025.
>
> [2] Loew et al. Universal machine learning interatomic potentials
> are ready for phonons. npj Computational Materials, 2025.
>
> [3] Barroso-Luque et al. OMat24. arXiv, 2024.

---

> > ### Author Rebuttal · Reviewer_w5RP · 2026-04-02
> >
> > Thank you for the detailed rebuttal. I appreciate the additional evaluations and clarifications. However, my main concerns are only partially addressed. First, the practicality issue remains: the method relies on DFT force constants, and the rebuttal mainly explains that such data are valuable, rather than showing that the approach is broadly applicable in realistic fine-tuning settings. Second, co-training appears to be more of a workflow-dependent design choice than a core part of the method itself, as it is mainly introduced to preserve upstream performance. Third, the current experiments still mainly support an advantage on the KSRME-related metrics, while the practical impact of the observed degradation on the other evaluation metrics has not yet been fully established. Overall, I will keep my score unchanged.

---

> > > ### Author Response · Authors · 2026-04-02
> > >
> > > Thank you for taking the time to elaborate your remaining concerns. We hope to address these with additional clarifications below:
> > >
> > > **1. Practicality/applicability**. We would like to clarify that we do not claim that PFT is the right fine-tuning strategy for all MLIP use cases, but a targeted procedure for workflows where accurate phonon and thermal properties are important. In practice it is already common to run DFT calculations to generate task-specific data to fine-tune existing foundation models. When phonon and thermal properties matter, this work helps inform *which DFT calculations are most useful to perform*; instead of relying only on standard energy/force/stress calculations, we show that one can benefit substantially from phonon calculations that provide force-constant information. As discussed in Sec. 2.3, these labels are commonly obtained from standard finite-displacement phonon calculations using widely used tools such as phonopy [1] together with conventional DFT software. Thus, PFT does not require a non-standard data generation workflow, and leverages labels that are already produced in phonon calculation workflows where practitioners need to compute vibrational properties.
> > >
> > > **2. Cotraining**. We agree that co-training is introduced primarily to preserve upstream performance, and we do not view it as the main novelty of the paper. Phonon calculations at equilibrium represent only a subset of the full distribution of atomic configurations that foundation MLIPs are expected to handle. Because these data are concentrated near equilibrium, where force and stress labels are $\approx0$, fine-tuning only on phonon data can degrade performance on the rest of the configuration distribution, including non-equilibrium structures that remain important for downstream workflows. Co-training is therefore intended to reincorporate this broader distribution during fine-tuning. We view it as an important practical component of making PFT usable in practice when one wants to improve phonon-related properties without sacrificing performance on other tasks such as relaxation or stability prediction (Table 3).
> > >
> > > **3. Practical impact beyond $\kappa_{\mathrm{SRME}}$**. While the main highlights of this work are the analysis on Hessian errors and the significant improvements in MDR Phonon metrics and thermal conductivity predictions, the remaining metrics in Matbench Discovery (RMSD, $R^2$, MAE, Prec, DAF, F1, Acc) are based on a practical workflow that simulates high-throughput discovery of stable inorganic crystals [2]. Measuring the direct practical impact of changes in these global metrics is non-trivial and depends on the downstream task. However, one can still gain intuition by comparing the relative differences between models on these metrics on the current leaderboard [2]. From this perspective, the deviations between the co-trained PFT and non-PFT variants are substantially smaller than differences induced by other design choices such as parameter count or architecture. The main takeaway from these results is that the co-trained PFT models achieve substantial gains on the target phonon and thermal conductivity tasks while leaving these broader workflow metrics nearly unchanged.
> > >
> > > We hope these clarifications better convey the intended scope of our work. The focus of PFT is for applications where vibrational and transport properties matter, which as discussed in our rebuttal, is important, practically relevant, and a significant use case of existing MLIPs. Co-training is included to preserve utility in standard downstream materials workflows.
> > >
> > > [1] Togo et al. Implementation strategies in phonopy and phono3py. Journal of Physics: Condensed Matter, 2023.
> > >
> > > [2] Riebesell et al. A framework to evaluate machine learning crystal stability predictions. Nature Machine Intelligence, 2025.
> > >
> > > [3] Fu et al. Learning smooth and expressive interatomic potentials for physical property prediction, 2025.

---

### Official Review · Reviewer_mGf7 · 2026-03-12

**Soundness:** 3
**Presentation:** 2
**Significance:** 2
**Originality:** 3
**Overall Recommendation:** 4
**Confidence:** 4

**Summary:**

Universal MLIPs trained on energy, force, and stress objectives systematically misfit the curvature of the potential energy surface, and this curvature error degrades phonon-derived thermodynamic predictions. PFT addresses this by adding a Hessian supervision term to the training loss, matching MLIP second derivatives to DFT-computed force constants. To keep training tractable on large supercells, it samples single Hessian columns and computes the loss via a Hessian-vector product, reducing per-step cost from O(N^2) to O(N). A co-training scheme interleaves upstream EFS data to prevent catastrophic forgetting, and the resulting method achieves state-of-the-art phonon and thermal conductivity accuracy among MPtrj-trained models.

**Compliance With Llm Reviewing Policy:**

Affirmed.

**Key Questions For Authors:**

1.Table A.2 marks lamda_phi = 100 as untuned. The entire gain in Table 1 rests on this term dominating the loss. Even a three-point sweep on the validation set would tell us whether the method is robust or happens to work well at this particular value. What does performance look like at lamda_phi = 10 and lamda_phi = 1000?

2.The claim that PFT generalizes to third-order derivatives is interesting but the evidence is indirect. Figure 5 is a cross-model correlation, not a controlled test of PFT's effect. Do the authors have third-order force constant errors before and after PFT for any subset of materials? If not, can they at least rule out that the K_SRME improvement comes entirely from better structural relaxation rather than improved curvature?

3.Section 4.1 sets force and stress labels to zero at supercell equilibrium. Given the energy discrepancy documented in Appendix A.2, some structures may have non-trivial residual forces after MDR's relaxation. What fraction of training structures have force magnitudes above, say, 5 meV/Å, and does excluding or reweighting them change the Table 1 results?

**Limitations:**

The JAX dependency is flagged in Section 6, and the note about phonon dataset diversity is present. Missing is any discussion of whether the zero-force assumption at supercell equilibrium introduces systematic bias, and there is no mention of expected performance on strongly anharmonic or disordered materials where harmonic force constants are less physically meaningful.

**Strengths And Weaknesses:**

Plotting Hessian MAE against phonon property error across six independent foundation models makes the motivation for direct Hessian supervision genuinely persuasive. The O(N) stochastic HVP strategy is well-justified, and Appendix A.1 backs it up with concrete numbers. Table 1 shows the 55% average MAE reduction cleanly, and the internal consistency is reassuring: the no-cotrain variant edges out the cotrain variant on phonon metrics (e.g., heat capacity MAE of 2 vs. 3 J/K/mol), which lines up with the small Hessian MAE penalty visible in Figure 3. The parity between finite-displacement and autodiff force constants in Table 1 also validates Section 3.5 without requiring a separate experiment.

The main soundness concern is the thermal conductivity claim in Section 4.3. The K_SRME improvement is real, but Figure 5 plots Hessian MAE against K_SRME across six architecturally diverse models trained on different data.The correlation there is largely driven by variation in model quality, not PFT specifically. The single PFT point in Figure 5 does not establish that second-order supervision causes third-order improvement. The mechanism is never explained, and no third-order force constant errors are reported before and after PFT.The second issue is lamda_phi = 100, which Table A.2 explicitly labels as not tuned. This is the loss term doing most of the work, and without a sensitivity analysis, it is unclear whether the Table 1 results are robust across reasonable values of lamda_phi or whether they depend on landing at a favorable operating point. Third, Section 4.1 assumes force and stress labels are zero at supercell equilibrium. Appendix A.2 documents a 31.60 meV/atom energy MAE between MDR Phonon and MPtrj, and some outliers reach 1.5 eV/atom. The same relaxation inconsistency could leave non-negligible residual forces in some training structures. This is noted nowhere in the main text.

---

> ### Author Rebuttal · Authors · 2026-03-27
>
> Thank you for the thorough evaluation of our work. We have included extra experiments to address every weakness (W) and question (Q) below. If you feel our response addresses your concerns, we kindly ask that you consider whether it merits an increase in score.
>
> **Q1/W2: $\lambda_\Phi$ Sensitivity**: We have added Figure A.4 below, which sweeps $\lambda_\Phi$ (without co-training) for the suggested values of (10, 100, 1000). This demonstrates the expected tradeoff: increasing $\lambda_\Phi$ improves Hessian error at the expense of higher errors for the other properties; the inverse occurs for the lower $\lambda_\Phi$. The $\lambda_\Phi=10$ and $\lambda_\Phi=1000$ models achieve Table 1 phonon metrics ($\omega_{\max}$/$S$/$F$/$C_V$) of (17/32/11/4) and (10/9/3/2) respectively, both better than the base model, with the latter outperforming the original $\lambda_\Phi=100$ PFT model (12/14/5/3). This demonstrates that our method is robust to choice in hyperparameter and corroborates our other findings of Hessian error correlating with phonon metrics.
>
> To further validate the robustness of the hyperparameter, we have applied PFT to a larger pre-trained model with a different architecture, MACE-MP-0 (4.7M parameters). Using identical PFT hyperparameters to the Nequix MP PFT model, we find that we achieve comparable gains to the Nequix model, with a 77% average improvement in phonon metrics MAE, and a 38% improvement in $\kappa_\text{SRME}$, which we show in Revised Table 1 and Revised Table 2 below.
>
> **Q2/W1**: We acknowledge that the improvement in thermal conductivity predictions may be due to factors outside the accuracy of third-order force constants. To address this directly, we extracted the displacement data from [1], computed the ground truth DFT third-order force constants, $\Phi^{(3)}$, then computed the MAE for each model before and after PFT. This is shown in New Table 4 below. *We find that with all three base models (Nequix MP, MACE-MP-0, and Nequix OAM), there is a 20-30% reduction in MAE of the third-order force constants after applying PFT.* We will, however mention in the text that this may not be the sole contributor to the improvement in thermal conductivity predictions.
>
> **Q3/W3**: We note that the discrepancy in energy between the MDR structures and MPtrj may be due to factors unrelated to convergence of the relaxation. Slight differences such as VASP version, settings, or choice of pseudopotential could cause this deviation. We have reached out to the authors [2] for an explanation, but they have not responded. While the original force data is not supplied, it is written in [2]:
>
> >  For the stringent relaxation step, [...], we used a higher energy and force convergence criteria of 10$^{−8}$ eV/cell and 10$^{−8}$ eV/Å, respectively.
>
> Because we do not have access to the forces, we rely on the published relaxation criteria, and we believe it is reasonable to assume the forces are effectively 0 after relaxation. We will add a clarification in the paper regarding this.
>
>
> **New Figure A.4**: (https://anonymous.4open.science/r/pft-icml/figures/lambda_phi.png) Sweep of $\lambda_\Phi$ hyper-parameter of values (10, 100, 1000) for PFT with the Nequix MP model.
>
> **Revised Table 1**: Evaluation of models on held-out MDR Phonon data. Metrics are MAE of maximum phonon frequency $\omega_\text{max}$ (K), vibrational entropy $S$ (J/K/mol), Helmholtz free energy $F$ (kJ/mol), and heat capacity at constant volume $C_V$ (J/K/mol). *Old results not shown due to character limit; see Table 1 in submission for rest of the results*.
> | Model | $\omega_{\max}$ | $S$ | $F$ | $C_V$ |
> |---|---|---|---|---|
> | MACE-MP-0 | 61 | 60 | 24 | 13 |
> | ----
> | MACE-MP-0 PFT (no cotrain) | 19 | 14 | **4** | 3 |
>
>
>
> **Revised Table 2**: Matbench Discovery "compliant" leaderboard for thermal conductivity, measured in symmetric relative mean error in predicted phonon mode contributions to thermal conductivity $\kappa_{\mathrm{SRME}}$. *Old results not shown due to character limit; see Table 2 in submission for rest of the results*.
> | Model | Params | $\kappa_{\mathrm{SRME}}\downarrow$ |
> |---|---|---|
> | MACE-MP-0 | 4.69M | 0.638 |
> | ----
> | MACE-MP-0 PFT (no cotrain) | 4.69M | 0.397 |
>
> **New Table 4**: MAE of third order force constants $\Phi^{(3)}$, measured in $\mathrm{meV}/\text{Å}^3$.
> | Model | $\mathrm{MAE}(\Phi^{(3)})$ |
> |---|---|
> | Nequix MP | 10.52 |
> | Nequix MP PFT (no cotrain) | **7.46** |
> | Nequix MP PFT | 8.35 |
> | ----
> | Nequix OAM | 6.46 |
> | Nequix OAM PFT | **5.13** |
> | ----
> | MACE-MP-0 | 11.41 |
> | MACE-MP-0 PFT (no cotrain) | **7.86** |
>
> [1] Togo, A. Phono3py input data for 103 compounds calculated using the finite displacement method with PBE, 2025.
>
> [2] Loew et al. Universal machine learning interatomic potentials are ready for phonons. npj Computational Materials, 2025.

---

> > ### Author Rebuttal · Reviewer_mGf7 · 2026-04-04
> >
> > I have read the rebuttal. My original review posed questions to the authors, which the rebuttal has now addressed. I have posted my detailed post-rebuttal assessment as an Official Comment.

---

> > > ### Author Response · Authors · 2026-04-04
> > >
> > > Thank you for reading the rebuttal and for the update. We are glad that your questions have been addressed. We are currently unable to see any Official Comments; if there are any remaining questions or concerns we would be happy to address these.

---

### Decision · Program_Chairs · 2026-04-30

**Decision:**

Accept (regular)

**Comment:**

This paper studies an important and well-motivated problem: standard MLIP pretraining on energies, forces, and stresses does not necessarily yield an accurate characterization of the curvature of the potential energy surface, which in turn limits performance on phonon-related and thermal properties. The proposed phonon fine-tuning (PFT) framework addresses this issue by directly supervising force constants, while introducing stochastic Hessian-column sampling and Hessian-vector products (HVPs) to reduce the computational burden. Overall, the method is conceptually simple, physically well grounded, and empirically effective.

The reviewers generally agree on the importance of the problem and the empirical strength of the paper. Several concerns were raised during review, most of which have been adequately addressed in the rebuttal.

* On the causal connection between second-order supervision and higher-order / thermal improvements, as asked by Reviewers mGf7 and yQZH. The authors addressed this concern with direct empirical evidence showing that PFT reduces third-order force-constant errors across multiple base models.
* Reviewer mGf7 also questioned the robustness of the hyperparameter setting and the soundness of the zero-force assumption at supercell equilibrium. The authors strengthened the paper by adding a sensitivity analysis over $\lambda_{\phi}$, and by clarifying that the energy discrepancy between the MDR structures and MPtrj may stem from differences in VASP version, calculation settings, or pseudopotential choice. This explanation appears reasonable to me.
* Reviewer w5RP raised concerns about the applicability of the method beyond phonon-related tasks and beyond the particular MLIPs considered, as well as the practical impact beyond the $\kappa_{\rm RMSE}$ metric. The authors provided additional empirical evidence in the rebuttal that partially but meaningfully addresses these concerns.
* On the computational overhead, as raised by Reviewers yQZH and w5RP. The authors’ response is convincing to me, especially in light of the additional discussion of relative fine-tuning cost and the new cross-architecture results.
* On prediction accuracy for non-equilibrium structures, raised by Reviewer yQZH. Additional MD stability results provided in the discussion help mitigate this concern. I nevertheless encourage the authors to further clarify this point in the final version of the paper.

Overall, the issues seem minor relative to the contribution of the paper.